# The Unexplored Role of Connexin Hemichannels in Promoting Facioscapulohumeral Muscular Dystrophy Progression

**DOI:** 10.3390/ijms26010373

**Published:** 2025-01-04

**Authors:** Macarena Díaz-Ubilla, Mauricio A. Retamal

**Affiliations:** 1Instituto de Ciencias e Innovación en Medicina, Facultad de Medicina, Clínica Alemana Universidad del Desarrollo, Santiago 7550000, Chile; 2Programa de Comunicación Celular en Cáncer, Facultad de Medicina, Clínica Alemana Universidad del Desarrollo, Santiago 7550000, Chile

**Keywords:** muscular dystrophy, therapy, connexin, hemichannels

## Abstract

DUX4 is typically a repressed transcription factor, but its aberrant activation in Facioscapulohumeral Muscular Dystrophy (FSHD) leads to cell death by disrupting muscle homeostasis. This disruption affects crucial processes such as myogenesis, sarcolemma integrity, gene regulation, oxidative stress, immune response, and many other biological pathways. Notably, these disrupted processes have been associated, in other pathological contexts, with the presence of connexin (Cx) hemichannels—transmembrane structures that mediate communication between the intracellular and extracellular environments. Thus, hemichannels have been implicated in skeletal muscle atrophy, as observed in human biopsies and animal models of Duchenne Muscular Dystrophy, Becker Muscular Dystrophy, and Dysferlinopathies, suggesting a potentially shared mechanism of muscle atrophy that has not yet been explored in FSHD. Despite various therapeutic strategies proposed to manage FSHD, no treatment or cure is currently available. This review summarizes the current understanding of the mechanisms underlying FSHD progression, with a focus on hormones, inflammation, reactive oxygen species (ROS), and mitochondrial function. Additionally, it explores the potential of targeting hemichannels as a therapeutic strategy to slow disease progression by preventing the spread of pathogenic factors between muscle cells.

## 1. Introduction

Facioscapulohumeral Muscular Dystrophy (FSHD) is one of the most prevalent types of muscular dystrophy, affecting between 1 in 8000 to 1 in 22,000 people worldwide [1,2,3]. This condition is caused by a genetic mutation and is characterized by a slow, progressive, and asymmetric weakening of specific muscle groups, including the muscles of the face, shoulder girdle, arms, abdominal wall, hips, and thighs [2,4]. The clinical presentation of FSHD is highly variable, even among individuals with the same genetic background, such as monozygotic twins, where differences in disease severity and muscle involvement have been documented [5,6]. This variability extends to the pattern of disease progression, which is often unpredictable and varies widely from person to person [3,7,8]. The unpredictable nature of FSHD likely reflects underlying uncertainties at the molecular level, particularly the factors governing the expression of the disease-causing gene, DUX4. Although DUX4 is a critical factor in the pathogenesis of FSHD, its expression is rare, detected in only 1 out of 1000 FSHD myoblasts and in approximately 1 out of 200 myotube nuclei during in vitro studies [9]. Despite extensive research, the DUX4 protein has not been successfully detected in muscle biopsies from FSHD patients [9,10]. However, even with such low levels of DUX4 expression, muscle atrophy continues to progress, affecting both initially impacted and surrounding healthy muscle fibers and eventually spreading to additional muscle groups. This suggests that FSHD involves a broader mechanism of pathogenic factor transmission between muscle cells, which accelerates muscle damage. One possible mechanism behind this transmission involves connexin (Cx) hemichannels, which facilitate the exchange of signaling molecules between the cell cytoplasm and the extracellular environment [11,12,13]. Cx hemichannels play a crucial role in intercellular communication [11,12,13], and their de novo expression has been linked to skeletal muscle atrophy in several muscular dystrophies, including Duchenne Muscular Dystrophy [14], Becker’s Muscular Dystrophy [14], and Dysferlinopathies [15]. These findings suggest that Cx hemichannels may represent a shared mechanism contributing to muscle degeneration across different types of dystrophies. However, this potential role of Cx hemichannels in FSHD has not yet been fully investigated or understood. This review aims to delve into the mechanisms that drive the progression of FSHD, with a particular focus on the role that Cx hemichannels may play in muscle atrophy. By examining these channels, we can explore their possible involvement in transmitting pathogenic factors from one muscle cell to another, contributing to the spread of disease. In addition, this review will discuss emerging therapeutic strategies that target Cx hemichannels and could slow the progression of FSHD by disrupting the intercellular communication of harmful signals. By preventing the spread of these pathogenic factors, hemichannel-targeted treatments could offer a promising approach to alleviating muscle damage and preserving muscle function in individuals with FSHD.

## 2. Facioscapulohumeral Muscular Dystrophy

Facioscapulohumeral Muscular Dystrophy (FSHD) is an autosomal dominant disorder, typically manifesting in late adolescence or early adulthood [3,16]. While it can be diagnosed at any stage of life [3], it affects men and women equally, although women tend to have a milder phenotype and later onset compared to men [17,18]. Over 95% of patients develop the classical form of the disease, known as FSHD type 1 (FSHD1, OMIM #158900), which is caused by the contraction of the D4Z4 tandem-repeat array located in the subtelomeric region of chromosome 4q35 [19,20,21]. Healthy individuals possess 11 to 200 D4Z4 repeats, while FSHD1 patients have only 1 to 10. This reduction in repeats results in hypomethylation of the region [22,23], leading to DNA unwinding and the de-repression of the DUX4 gene [24,25,26] (Figure 1A). Besides the genetic complexity of FSHD, there is a growing field unveiling several epigenetic mechanisms promoting DUX4 expression. In vitro studies conducted on cells derived from FSHD patients have reported a reduction in the enrichment of the repressive mark H3K9me3 at the D4Z4 array [27]. The shortening of the D4Z4 region also promotes the expression of the DBE-T long non-coding RNA (“D4Z4 binding element”), the most studied lncRNA in FSHD. It has been detected in both primary cell cultures from patients and in biopsies [28]. To date, DBE-T has been proposed as a master regulator of DUX4, as it contributes to its transcriptional expression by recruiting ASH1L, a chromatin-activating protein belonging to the Trx group [28,29]. This facilitates the transition from heterochromatin to euchromatin [29]. Another study confirmed the dysregulation of epigenetic pathways involving ASH1L, BRD2, KDM4C, and SMARC5 in primary myocytes from FSHD1 patients. Silencing these pathways resulted in decreased levels of the DUX4-fl transcript and its transcriptional targets [30]. Interestingly, while epigenetic mechanisms like those mentioned above contribute to DUX4 expression—and thus to the manifestation of the FSHD phenotype—DUX4 expression itself, in turn, promotes epigenetic changes. A recent study demonstrated that DUX4 induces the expression of histone variants H3.X and H3.Y, which incorporate into the target gene regions of DUX4 [31]. Previous biochemical analyses have shown that nucleosomes containing H3.Y exhibit a more relaxed chromatin configuration compared to those formed with histone H3.3 [32]. This suggests that the H3.Y variant, and likely H3.X as well, incorporates into transcriptionally active genes and helps maintain an accessible chromatin conformation [31]. This observation is relevant because it partially helps explain how the low and sporadic expression of DUX4 is sufficiently prolonged and amplified to trigger the clinical phenotype.

A small subset of patients develops FSHD type 2 (FSHD2, OMIM #158901), which has an identical clinical presentation to FSHD1. Despite FSHD2 being caused by mutations in the SMCHD1 gene rather than a contraction of the D4Z4 array, both conditions ultimately result in higher DUX4 expression (or Cx function) [3]. During early pre-implantation development, DUX4 is essential for zygotic genome activation [33,34,35]. After this stage, DUX4 is silenced in all adult tissues except the testis, thymus, and keratinocytes [16,33]. However, in FSHD, the inappropriate activation of DUX4 in skeletal muscle—due to the complex genetic and epigenetic mechanisms mentioned above—leads to cell death by disrupting cell homeostasis. This occurs through the alteration of several pathways, including impaired myogenesis, compromised sarcolemma integrity, dysregulated gene expression, oxidative stress, altered immune responses, and other disrupted biological processes [22,26,36,37,38,39,40] (Figure 1B).

### FSHD Progression

Phenotypically, the progression of FSHD is slow, highly variable between individuals, and follows an asymmetrical pattern, which makes studying the disease particularly challenging. Interestingly, FSHD often progresses from the upper to lower extremities in a cranio-caudal direction [16,41,42] (Figure 2). At the molecular level, the mechanisms driving FSHD progression are not fully understood and have not been directly correlated with the shortening of the D4Z4 tandem repeats. However, recent research has shown a moderate correlation between reduced D4Z4 methylation and the severity of patient phenotypes [43]. This reduction in methylation has even been proposed as a potential diagnostic tool, as it effectively distinguishes between healthy individuals, FSHD1, and FSHD2 phenotypes [44]. Regarding DUX4, studies have demonstrated a progressive increase in its levels, activity, and mRNA expression during myoblast differentiation into myotubes [45]. Both conditions ultimately result in higher DUX4 expression. Additionally, treatment with estrogen has been shown to reduce DUX4’s nuclear localization and transcriptional activity via estrogen receptor β (ERβ), which antagonizes DUX4’s DNA-binding capacity [17]. This molecular observation is supported by a study showing that patients with FSHD who received estradiol significantly downregulated DUX4 [46]. Another study analyzing hormone levels in FSHD patients’ peripheral blood concluded that higher estradiol and progesterone levels, as well as favorable ratios of these hormones to testosterone, might be associated with milder disease severity [47]. These findings suggest that female hormones may play a protective role in slowing FSHD progression, offering a possible explanation for why women tend to have a milder phenotype.

A key hallmark of FSHD is muscle inflammation [48,49,50]. Analysis of mononuclear cells in muscle sections has shown a strong correlation between the number of inflammatory cells and the extent of necrotic muscle fibers in FSHD patients [51]. A recent study using RNA deconvolution techniques on FSHD muscle biopsies revealed a higher relative contribution of myeloid cells compared to healthy controls [52]. Longitudinal studies have also demonstrated that edema, indicative of active inflammation as seen on quantitative MRI (STIR hyperintensity), often precedes the replacement of muscle tissue with fat [53,54]. The authors of these studies suggest that muscle inflammation accelerates muscle degradation and continues until the affected muscles are completely replaced by fat [54]. Another study using T2-STIR imaging detected inflammatory infiltrates in FSHD patients, primarily composed of CD8+ T cells within the endomysium. These findings correlated with the presence of activated immune cells—mainly CD8+ T cells—in the peripheral blood of FSHD patients [55]. Additionally, a study analyzing 20 different cytokines in 100 FSHD patients found that IL-6 levels were more than double those observed in healthy controls [56], further highlighting the role of inflammation in FSHD pathology. A significant correlation was found between IL-6 levels and three well-established clinical severity and functional scores: MMTsum core, Brooke score, Vignos score, and CSS [56]. Although the role of IL-6 in neuromuscular pathologies is not fully understood, elevated IL-6 levels have been used as a disease progression biomarker in certain groups of amyotrophic lateral sclerosis (ALS) patients, specifically those with the IL6R358Ala variant [57]. Collectively, this evidence strongly suggests that inflammation precedes muscle atrophy and fibrofatty degeneration in FSHD, contributing to disease progression.

Research has also focused on the roles of reactive oxygen species (ROS) and mitochondrial dysfunction in FSHD. Increased ROS production has been reported in FSHD myoblasts [58], likely due to mitochondrial impairment. This dysfunction has been linked to reduced mitochondrial biogenesis during FSHD myogenesis, attributed to the suppression of two key proteins involved in mitochondrial biogenesis: PGC1α and ERRα [42]. Interestingly, this effect was reversed when FSHD myoblasts were treated with phytoestrogens such as biochanin A, genistein, or daidzein (all ERRα agonists), which promoted healthy myogenesis. The protective effects of phytoestrogens suggest they may slow FSHD progression by restoring normal muscle development. A similar effect was observed in a clinical trial involving antioxidant supplementation. In a randomized, placebo-controlled trial, supplementation with 500 mg of vitamin C, 400 mg of vitamin E, 25 mg of zinc, and 200 µg of selenium significantly improved maximal voluntary contraction of both quadriceps. This improvement was attributed to enhanced antioxidant defenses and reduced oxidative stress in FSHD patients [59]. Thus, in addition to female hormones, phytoestrogens, and antioxidants may offer therapeutic potential in reducing disease progression in FSHD by targeting mitochondrial health and supporting proper myogenesis.

## 3. Connexin Hemichannels

Cxs are membrane proteins expressed exclusively in vertebrates and encoded by 21 different genes in humans [60]. Each Cx consists of four transmembrane domains, two extracellular loops, and one intracellular loop [61]. After synthesis, Cxs oligomerize in the endoplasmic reticulum (ER) and/or the trans-Golgi apparatus, forming hexamers known as hemichannels [62,63]. These Cx hemichannels are then transported to the plasma membrane via vesicles that move along the cytoskeleton [64]. Once at the plasma membrane, Cx hemichannels typically remain closed to prevent cell lysis [13], as they are permeable to ions like Ca^2+^ and Na^+^ and small molecules such as ATP, glutathione, glutamate, and glucose [65,66,67,68,69,70,71]. Because of this permeability, Cx hemichannels can open in a controlled manner, allowing the release of small but biologically significant signaling molecules into the extracellular space. For instance, Contreras et al. demonstrated that Cx43 hemichannels exhibit hemichannel-mediated currents under normal conditions, with a low but nonzero frequency of opening [72]. Additionally, dye uptake studies consistently show that cells expressing Cxs display basal dye uptake, indicating ongoing, low-level hemichannel activity [72,73,74]. However, in various diseases, Cx hemichannel activity increases significantly [75,76], leading to cell damage by allowing excessive Ca^2+^ uptake [77] and the release lysis of important metabolites [75]. In summary, Cx hemichannels at the plasma membrane act as low-probability channels, capable of releasing and uptaking key signaling molecules, thereby influencing neighboring cells (Figure 3).

When two Cx hemichannels from adjacent cells dock together, they form a gap junction channel (GJC), facilitating direct cell-to-cell communication [78]. GJCs are crucial for the transfer of small metabolites, second messengers, and electrical signals between neighboring cells [78]. This communication enables both electrical and metabolic coordination, ensuring proper tissue and organ function. A notable example is seen in cardiomyocytes, which, unlike skeletal muscle, express Cx43 in adulthood. This Cx forms GJCs between cardiomyocytes, allowing the synchronized spread of electrical impulses necessary for coordinated heart muscle contraction [79,80]. However, under certain pathological conditions, such as ischemic heart failure, the number of Cx43 GJCs tends to decrease while the number of active Cx hemichannels at the plasma membrane increases. This imbalance disrupts electrical signaling, leading to arrhythmias. Interestingly, this pattern of increased hemichannel activity is observed not only in cardiac muscle but also in skeletal muscle under pathological conditions. Studies have shown that blocking hemichannels, particularly with Cx43 hemichannel mimetic peptides, can improve cardiac function by preventing electrical disturbances and restoring proper cell communication [81,82,83]. This highlights the potential of hemichannel blockers as a therapeutic strategy for conditions affecting both cardiac and skeletal muscles.

### Hemichannels in Skeletal Muscle Atrophy

In skeletal muscle cells, Cx GJCs are essential during myogenesis, particularly Cx43 GJCs [84]. In addition to Cx43, Cx39, Cx40, and Cx45 have been detected in developing myoblasts (skeletal muscle precursor cells) [85]. As myoblasts undergo fusion to form mature muscle fibers, these Cx channels are downregulated until no detectable Cx expression remains in adult muscle fibers [85,86,87]. In fact, while Cx is vital in most body tissues, adult skeletal muscle is one of the few tissues where Cxs are absent [85,86,87]. Cx hemichannels play a significant role in the homeostatic imbalances seen in various diseases [75]. Enhanced Cx hemichannel activity is known to induce or accelerate cell death in several pathological conditions [75,88], including in muscle fibers. Notably, de novo Cx expression and hemichannel activity have been associated with skeletal muscle atrophy in human biopsies and animal models of Duchenne Muscular Dystrophy [14], Becker’s Muscular Dystrophy [14], and Dysferlinopathies [15]. This suggests that Cx involvement in muscle degeneration is independent of the specific genetic mutations underlying these dystrophies. Moreover, muscle atrophy resulting from other causes—such as chemical induction [89], denervation [90,91], and endotoxemia [92]—has also been linked to the de novo expression of Cx43- and Cx45-based hemichannels. Blocking these Cx hemichannels has been shown to prevent muscle atrophy, as well as mitochondrial dysfunction, inflammation, and oxidative stress [91,93]—processes closely related to the progression of Facioscapulohumeral Muscular Dystrophy (FSHD).

Since FSHD is driven by the aberrant expression of the DUX4 transcription factor, a critical question is whether Cx genes are direct or indirect targets of DUX4’s transcriptional activity. Current evidence suggests they are not. Skeletal muscle cell death in other pathologies occurs independently of DUX4, and no studies have demonstrated DUX4-mediated regulation of *Cx* genes in muscle or other tissues. However, an indirect effect remains plausible, as DUX4 induces oxidative stress and inflammation—conditions known to alter Cx expression and function [94,95,96,97]. For instance, Cx43 hemichannels are often dysregulated under inflammatory conditions or oxidative stress [71,98,99]. DUX4 also modulates the cellular microenvironment by regulating genes involved in pathways such as inflammation and autophagy, which could indirectly influence Cx expression and function. For example, increased DUX4 expression has been correlated with elevated heme oxygenase-1 (HO-1) levels in hypoxia-resistant cells, and HO-1 activity has been shown to modulate Cx43 and Cx46 hemichannel activity [100]. Regardless of whether DUX4 directly or indirectly regulates *Cx* expression and function, exaggerated hemichannel activity in dystrophic muscle fibers may increase sarcolemma permeability to critical metabolites like ATP and Ca^2+^. This dysregulation could contribute to cellular dysfunction or death and facilitate the transfer of atrophy-inducing factors—such as calcium ions, free radicals, or extracellular vesicles—between muscle fibers. While sarcolemmal reorganization has been observed in FSHD, its potential association with functional hemichannels remains unexplored, underscoring the need for further investigation (Figure 4).

## 4. Targeting Hemichannels: A Novel Therapeutic Avenue for FSHD

Despite numerous proposed therapeutic strategies, no effective treatment or cure currently exists for FSHD patients [19,101,102,103]. Since the identification of DUX4 as the primary driver of muscle degeneration in FSHD, most therapeutic approaches have focused on targeting this protein. Efforts to silence DUX4 at the transcriptomic, translational, or epigenetic levels—through CRISPR technology or small molecules—represent the most ambitious strategies [19,101,102,103]. However, the considerable variability in patient phenotypes (ranging from asymptomatic carriers to individuals with chronic respiratory failure, influenced by genetic, epigenetic, and gender factors), the lack of suitable animal models, and the absence of reliable biomarkers pose significant challenges to both preclinical and clinical research [39,41,104,105,106]. In this context, the pursuit of treatments aimed at slowing disease progression or preventing the spread of muscle toxicity is becoming increasingly relevant, particularly for early-onset patients. Children diagnosed with FSHD tend to experience more severe disease phenotypes—about 40% of those with infantile FSHD become wheelchair-dependent by an average age of 17, compared to 10% of the general FSHD population [104]. Notably, studies in a dysferlinopathy mouse model demonstrated muscle and systemic protection when myofibers were deficient in Cx43 and Cx45 expression, suggesting that muscle atrophy mechanisms are downstream of these Cxs [107]. Furthermore, inhibiting Cx43 and Cx45 hemichannels has shown significant protective effects in animal models of dysferlinopathy and Duchenne muscular dystrophy [15]. These findings imply that increased sarcolemmal permeability due to Cx hemichannel activity may be an early trigger of muscle atrophy. Cytokine-induced sarcolemmal permeability via hemichannels has also been reported, making inflammation another factor that exacerbates FSHD progression [92]. Thus, blocking Cx hemichannels before muscle loss or its spread could be crucial in slowing disease progression and preserving muscle mass and strength in FSHD patients (Figure 5). In a clinical trial involving FSHD patients, an antioxidant cocktail that included vitamin E led to improved muscle strength [59]. Interestingly, in 2020, vitamin E was found to block Cx43 and Cx45 hemichannels in vitro and in vivo, an effect linked to the prevention of muscle atrophy, mitochondrial dysfunction, and oxidative stress [93].

On the other hand, Sáez and collaborators have conducted extensive research into the effects of boldine on various muscle diseases, highlighting its therapeutic potential in addressing conditions characterized by oxidative stress and inflammation [108]. Boldine, an alkaloid derived from the Chilean tree *Peumus boldus*, has gained attention due to its antioxidant and anti-inflammatory properties, which have been shown to mitigate muscle damage in several disease contexts. However, boldine’s therapeutic effects extend beyond its well-known antioxidant activity. Recent studies suggest that boldine also acts as a Cx hemichannel blocker [91,109], though the precise mechanism underlying this action remains unclear. Notably, this blocking effect appears to be independent of boldine’s antioxidant properties [110], indicating a potential direct interaction with Cx hemichannels that warrants further investigation. Evidence increasingly supports boldine’s role as a protective agent against the deleterious effects of muscular dystrophies, particularly dysferlinopathy. In muscle cell cultures derived from dysferlinopathy patients, boldine exposure shifted the differentiation pathway of the cells from adipogenic to myogenic, promoting the formation of muscle fibers rather than fat tissue [111]. This myogenic differentiation was also accompanied by an increase in the fusion of myogenic precursor cells, a crucial process in muscle regeneration. Interestingly, this effect was associated with the downregulation of peroxisome proliferator-activated receptor gamma (PPARγ) [111], a key regulator of adipogenesis. By decreasing PPARγ expression, boldine may help counteract muscle degeneration, promoting muscle repair and growth [111]. Additionally, boldine has shown protective effects in other muscle-related conditions, such as diabetes-induced myofiber atrophy [112]. In primary cultures of myofibers exposed to high glucose concentrations, elevated hemichannel activity led to increased ethidium bromide uptake, a marker of cellular damage. Boldine effectively mitigated this damage by reducing hemichannel activity [112], suggesting that it may offer protection against glucose-induced muscle atrophy, which is a common complication in diabetic patients.

More recent studies have also explored boldine’s potential in nerve denervation models, which simulate conditions like peripheral neuropathy and spinal cord injury [113]. In a mouse model of nerve denervation, daily administration of boldine enhanced the evoked response of the tibialis anterior muscle. Furthermore, boldine reduced the expression of Cx43 and Cx45, which are implicated in the propagation of harmful signals leading to muscle degeneration [113]. This reduction in Cx expression suggests that boldine may help prevent the spread of degenerative signals between muscle fibers, thereby alleviating the detrimental effects of nerve denervation. In another study focusing on spinal cord injury, boldine prevented damage to the gastrocnemius muscle, as demonstrated by significant changes in both transcriptomic and metabolomic profiles [114]. Although boldine did not prevent muscle weight loss in this model, the observed molecular changes indicate that it may still offer important protective effects at the cellular level [114]. These findings underscore boldine’s potential as a therapeutic agent for neuromuscular conditions, particularly those involving nerve damage and muscle atrophy. Overall, the evidence suggests that boldine may offer a promising therapeutic approach for a range of neuromuscular diseases. Its dual action as both an antioxidant and a hemichannel blocker provides a unique advantage, as it may help mitigate multiple pathological processes simultaneously. Boldine’s ability to promote muscle regeneration, reduce atrophy, and protect against nerve damage makes it a valuable candidate for future clinical trials aimed at developing new treatments for muscular dystrophies, diabetes-related muscle atrophy, and nerve injury-associated muscle degeneration.

Probably the most specific Cx43 hemichannel inhibitors are the mimetic peptides. Thus, targeting Cx hemichannels with an antibody directed against Cx-extracellular loops may help improve muscle cell survival or promote myogenic differentiation. For example, modulating Cx43 hemichannels with an antibody in osteocytes has been shown to release ATP, which serves as a paracrine signal to inhibit the migration and invasion of cancer cells in bone metastasis [115]. Applying this concept to muscle cells, activating or inhibiting hemichannels could regulate the extracellular environment, promoting signaling pathways that prevent atrophy. Additionally, selective hemichannel blockers, such as the TAT-GAP19 [116] or TAT-L2 [117] peptides, could reduce hemichannel activity in muscle cells, preventing excessive ATP release or upload of Ca^2+^, thereby protecting muscle cells from further damage while maintaining essential signaling. Moreover, a peptide mimicking the C-terminal region of Cx43 that interacts with ZO-1 [118], known as αCT1, has been used in wound healing and cancer therapy [119,120,121]. This peptide could potentially be explored in muscle repair or atrophy prevention by modulating hemichannel activity, enhancing muscle cell regeneration, and reducing fibrosis. These strategies, focused on hemichannel modulation, offer exciting possibilities for developing new treatments to combat muscle atrophy by leveraging the complex intercellular communication mediated by hemichannels.

In summary, while antioxidants reduce ROS levels, protecting against cellular damage and improving mitochondrial function—which may help preserve muscle fiber integrity—some Cx-based mimetic peptides specifically inhibit hemichannel activity without disrupting gap junction-mediated intercellular communication. By doing so, they have the potential to reduce cell lysis and death while maintaining proper signaling flow between cells. Furthermore, these peptides can significantly decrease extracellular ATP levels and ion dysregulation, leading to reduced muscular inflammation and apoptosis. Their high specificity minimizes off-target effects, enhancing their potential as therapeutic agents. However, these approaches are not without limitations. The effectiveness of antioxidants depends on their ability to penetrate muscle tissues and achieve therapeutic concentrations without triggering pro-oxidant effects, which can arise when ROS levels are excessively suppressed [122,123]. Additionally, long-term antioxidant use may interfere with physiological redox signaling, a process essential for normal cellular function [122,123]. Mimetic peptides, on the other hand, face challenges related to stability, delivery, and bioavailability [124]. These peptides often require advanced delivery systems to reach their targets effectively, and their susceptibility to enzymatic degradation in the bloodstream can further reduce therapeutic efficacy. Moreover, the heterogeneity of FSHD pathophysiology means that these treatments may address only specific aspects of the disease, such as oxidative stress or hemichannel dysfunction, without addressing the underlying causes of muscle degeneration. Recognizing these limitations underscores the need for further research aimed at optimizing the delivery, efficacy, and safety of antioxidants and mimetic peptides. Combining these approaches with other therapeutic strategies could provide a more comprehensive and effective solution for managing the multifaceted nature of FSHD.

## 5. Discussion

Future research into therapies targeting hemichannels in FSHD should explore their potential roles beyond simply increasing sarcolemmal permeability. Emerging evidence suggests that hemichannels may also participate in the docking and fusion of extracellular vesicles, facilitating direct transmission of vesicular contents [125]. This alternative cell communication mechanism could further explain FSHD progression. Additionally, a recent study reported a significant increase in miR-206 expression in an FSHD animal model and in serum samples from FSHD patients [126]. Cx43 mRNA contains two binding sites for miR-206 in its 3′-untranslated region, both of which are necessary for efficient downregulation during myoblast differentiation [127]—a pathway known to be disrupted in FSHD. Unlike DUX4-targeted strategies, which would need to be tailored specifically to FSHD1 or FSHD2 (as these subtypes are coded in different genetic loci), therapies targeting Cx hemichannels could be effective in both forms of the disease. Hemichannel activity is hypothesized to be a common contributor to muscle atrophy, regardless of the underlying cause. While the onset and clinical symptoms of FSHD may vary by sex and other factors, all patients face disease progression. Targeting hemichannels to slow this progression could be life-changing for many patients, especially as genetic therapies or a potential cure are still on the horizon. One compelling reason to investigate hemichannel blockade as a therapeutic strategy in FSHD is that many blocking agents, such as antioxidants, are well-characterized, with established safety profiles from previous clinical trials. These treatments are also considerably more affordable than advanced gene therapies, making them accessible to patients in low- and middle-income countries. Given their safety record, a clinical trial using such agents could gain approval more easily, potentially fast-tracking the process with a Phase I/II trial.

## 6. Conclusions

Future research on therapies targeting hemichannels in FSHD should expand beyond their direct role in cell-to-cell communication and investigate their broader contributions to disease progression, such as their involvement in extracellular vesicle transmission [128]. Emerging evidence suggests that hemichannels may facilitate the release of pathogenic factors through vesicles, further promoting muscle damage and atrophy. Therefore, a deeper understanding of these processes could open new therapeutic avenues. One potential regulatory pathway that should be explored in relation to Cx43 hemichannels is the influence of microRNA-206 (miR-206) [129]. This miRNA, which is known to play a key role in muscle regeneration, is disrupted in FSHD [130]. Its connection to hemichannel regulation may provide insights into how these channels contribute to the muscle degeneration seen in both FSHD1 and FSHD2. Targeting miR-206-associated pathways could complement hemichannel-focused therapies, offering a multifaceted approach to FSHD treatment. Unlike therapies that exclusively target DUX4—the primary genetic culprit in FSHD—hemichannel-targeting strategies hold the potential to be effective in both FSHD1 and FSHD2, as they address mechanisms that are not solely dependent on DUX4 expression. This broader applicability makes hemichannel therapies a promising alternative or adjunct to DUX4-specific treatments, particularly given the variability and unpredictability of DUX4 expression in FSHD muscle cells. Additionally, we propose that blocking agents, such as mimetic peptides and antioxidants, may provide a safe and cost-effective alternative to more complex gene therapies, particularly for patients in low- and middle-income countries. These agents, by inhibiting hemichannel activity, could mitigate muscle damage and slow disease progression. Mimetic peptides, which selectively inhibit hemichannels, have already shown promise in preclinical models of other diseases, and their application to FSHD could be similarly beneficial. Antioxidants, known for their ability to reduce oxidative stress—a key contributor to muscle degeneration—could further enhance the efficacy of hemichannel inhibitors. These treatment strategies could accelerate the path to clinical trials, offering an accessible and scalable therapeutic option while more advanced gene therapies are still under development. By providing an interim solution, hemichannel-targeting therapies may help alleviate the burden of FSHD in patients around the world, particularly in regions where access to cutting-edge genetic treatments may be limited.

## Figures and Tables

**Figure 1 ijms-26-00373-f001:**
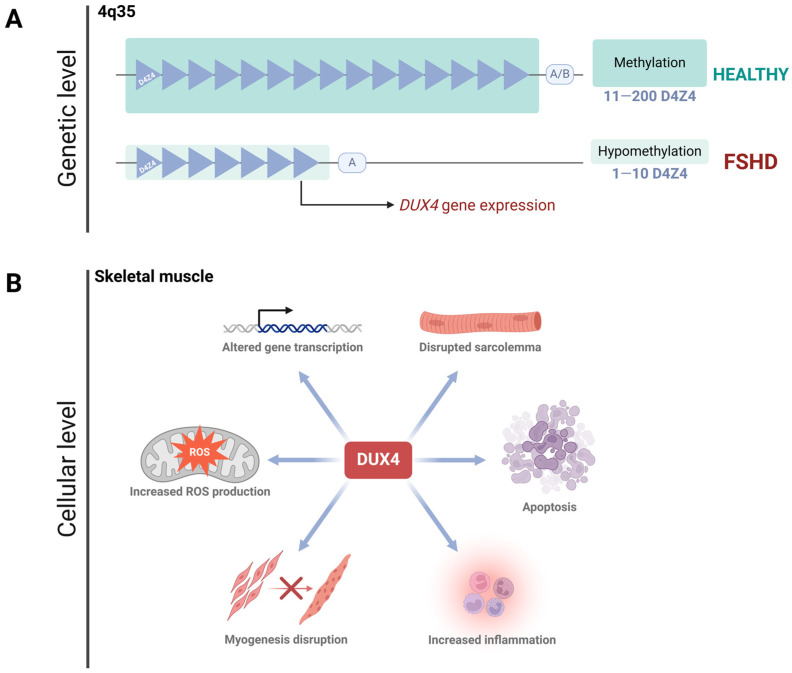
**FSHD: from DNA to cell homeostasis loss.** (**A**). FSHD type I is encoded in the 4q35 region, where a D4Z4 satellite contraction (less than 11 D4Z4 subunits left) promotes hypomethylation of the zone, allowing DUX4 expression. DUX4 expression requires the permissive allele A, while the non-permissive allele B prevents its expression. (**B**). DUX4 protein disrupts cell homeostasis in skeletal muscle. Since this protein is a transcription factor, its aberrant activity deregulates several pathways, including myogenesis, ROS production, gene transcription, sarcolemma integrity, inflammation, and apoptosis.

**Figure 2 ijms-26-00373-f002:**
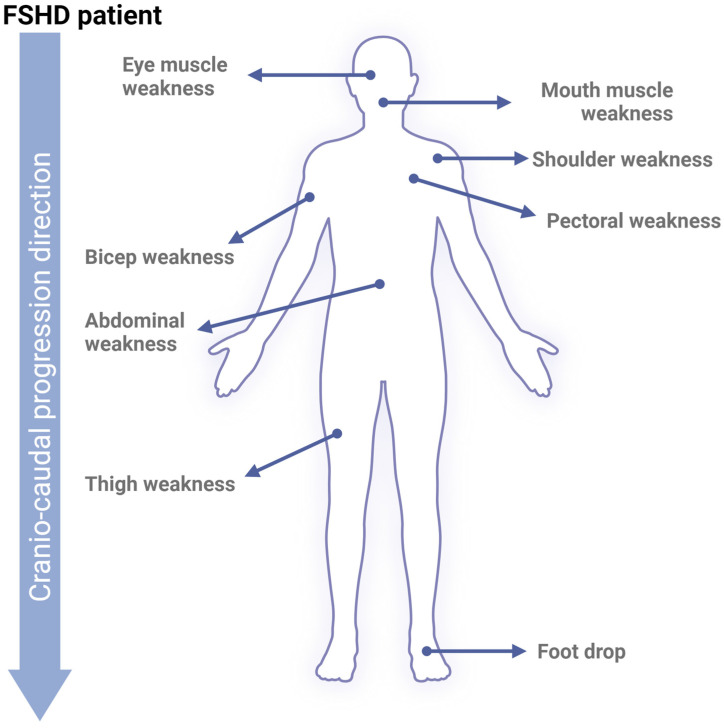
**FSHD phenotype.** Upon skeletal muscle cell apoptosis, muscle volume diminishes progressively, reducing strength and movement range. Early affected FSHD muscles are located in the face (around the eyes and mouth), shoulders, and upper arms. Following cranio-caudal progression, FSHD will later affect core, leg, and foot muscles.

**Figure 3 ijms-26-00373-f003:**
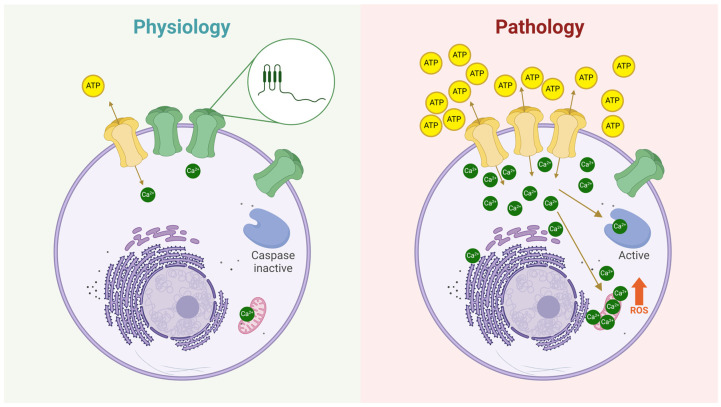
**The role of hemichannels in both physiological and pathological conditions.** Under physiological conditions (**left** panel), hemichannels at the plasma membrane exhibit a very low open probability (green hemichannels). However, some hemichannels can transiently open (yellow hemichannel), allowing the controlled release of signaling molecules such as ATP, glutathione (GSH), glucose, glutamate, amino acids, and others, or the uptake of ions like Ca^2+^ and Na^+^. The zoomed-in circle illustrates the structure of a Cx subunit, highlighting its topology at the plasma membrane: two extracellular loops, four transmembrane domains, an intracellular loop, and both N- and C-termini oriented toward the cytoplasm. In contrast, the **right** panel depicts hemichannels under pathological conditions, where their number and/or open probability is significantly increased. As a result, more hemichannels facilitate the release of metabolites (e.g., ATP) and the uptake of ions (e.g., Ca^2+^), leading to cellular dysregulation. These changes can activate apoptotic pathways, such as those mediated by caspase activation and reactive oxygen species (ROS) production.

**Figure 4 ijms-26-00373-f004:**
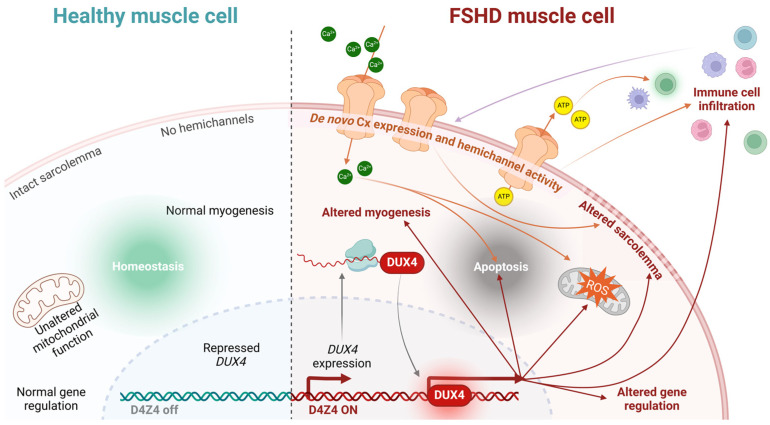
**Conceptual model linking hemichannel activity to FSHD muscular cell death.** Healthy muscle cells maintain homeostasis due to the proper functioning of their mitochondria, intact sarcolemma, absence of hemichannels, and normal gene expression (among other factors), thereby allowing myogenesis (**left** panel). Conversely, FSHD muscle cells expressing DUX4 exhibit a series of dysregulated pathways that eventually lead to apoptosis (**right** panel). De novo Cx expression and hemichannel activity in the cell membrane may promote the cell death process by directly altering the sarcolemma and increasing the intracellular concentration of ions, such as Ca^2+^, which contributes to apoptosis and elevated ROS levels. Hemichannel-mediated ATP release can activate immune cells, further promoting immune cell infiltration, which, in turn, modulates hemichannel activity through cytokine release.

**Figure 5 ijms-26-00373-f005:**
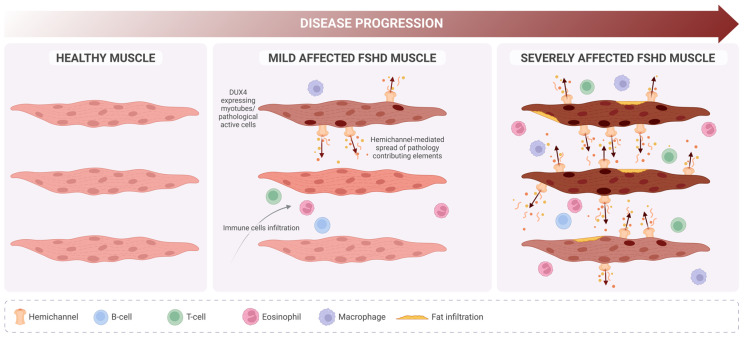
**Proposed role of hemichannels in FSHD progression.** In contrast with healthy muscle (**left**), in FSHD, mildly affected muscle (**middle**) DUX4 expression is detectable but only in a few myotubes (darker nucleus), which will undergo deregulation of several biological pathways. Pathology-contributing elements (i.e., calcium ions, ATP, free radicals, extracellular vesicles) originating from those cells might spread (dark arrows) and reach new cells due to de novo expression of hemichannels, promoting a severe phenotype (**right**) characterized by high immune cell infiltration and muscle fat replacement.

## Data Availability

Any data will be available when requested.

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
