# Peer review of "The Unexplored Role of Connexin Hemichannels in Promoting Facioscapulohumeral Muscular Dystrophy Progression"

_ijms, 2025, doi:10.3390/ijms26010373_

Round 1
Reviewer 1 Report
Comments and Suggestions for Authors
Major objections:
1. Line 91: Provide more biochemical information of primate-specific transcription factor DUX4.
2. Fig. 2: Is the diaphragm not affected in FSHD as in other muscular dystrophies? Pectoral weakness is too imprecise.
Minor objections:
1. Line 79: write: both conditions ultimately result in higher DUX4 expression (or Cx function)
2. Fig. 1A: correct methilation
3. Line 202: Write Hemichannels in Skeletal Muscle Atrophy
4. Line 226/227: What is meant with ‘athrophy inducing factors‘?
5. Line 289: Why is the tibialis anterior muscle ‘a key indicator of muscle function’? Define or delete this description
6. Figue 3: I assume that the different cells between the muscle fibers represent infiltrating immune cells or replacing fat cells. Please define the different colors of the cells in the Figure legend.
7. Add the spaces before the square brackets containing the reference numbers
8. Delete the full stop in the title
Author Response
Major objections:
- Line 91 (ahora 116): Provide more biochemical information of primate-specific transcription factor DUX4.
R: We have added the following paragraph in order to add more specific information “The DUX gene family is exclusive to placental mammals, but DUX4 is specific to primates [42]. Phylogenetic analyses suggests that DUX4 originated from the ancestral DUXC gene (currently present in mammals but absent in primates) [33]. One mutation in the N-terminal DNA-binding domain changes arginine to glutamate, altering the target sequence. This change has been suggested to have co-evolved with species-specific transposable elements (TEs), since DUX4 has been demonstrated to regulate a subset of transposable elements TEs [46]. Surprisingly, and despite DUX4 expression being restricted to primates, DUX4-positive nuclei were detected in C2C12 cells (a mouse-derived cell line), following co-culture with immortalized FSHD myoblasts [47]. Moreover, DUX4 activation in one nucleus appears to spread to adjacent nuclei [48], strengthening the hypothesis of a potential mechanism for disease progression.”
- Fig. 2: Is the diaphragm not affected in FSHD as in other muscular dystrophies? Pectoral weakness is too imprecise.
R: Dear Reviewer, thank you for your valuable suggestion. With “Pectoral weakness” we refer to the group of pectoral skeletal muscles. These set of muscles -being the pectoralis major the bigger one- bind the front walls of the chest with the bones of the upper arm and shoulder. Regarding diaphragm, currently there’s no consensus about this muscle being involved in the typical FSHD phenotype, although some isolated case reports have been published describing FSHD patients with diaphragm dysfunction.
Minor objections:
- Line 79: write: both conditions ultimately result in higher DUX4expression (or Cx function)
R: Fixed as requested
- Fig. 1A: correct methylation
R: Fixed as requested
- Line 202: Write Hemichannels in Skeletal Muscle Atrophy
R: Fixed as requested
- Line 226/227: What is meant with ‘athrophy inducing factors‘?
R: This sentence was deleted
- Line 289: Why is the tibialis anterior muscle ‘a key indicator of muscle function’? Define or delete this description
R: This sentence was deleted
- Figue 3: I assume that the different cells between the muscle fibers represent infiltrating immune cells or replacing fat cells. Please define the different colors of the cells in the Figure legend.
R: Fixed as requested
- Add the spaces before the square brackets containing the reference numbers
R: Fixed as requested
- Delete the full stop in the title
R: Fixed as requested
Reviewer 2 Report
Comments and Suggestions for Authors
In this review, the authors examine the mechanisms driving FSHD progression, emphasizing the roles of hormones, inflammation, reactive oxygen species (ROS), and mitochondrial function. They also discuss the potential of hemichannel-targeted therapies to mitigate disease progression by inhibiting the transmission of pathogenic factors between muscle cells.
The article is a significant contribution to FSHD research, with a unique focus on connexin hemichannels. However, it could benefit from a deeper exploration of the direct connections between these mechanisms and the specific pathogenesis of FSHD, as well as a more critical discussion of therapeutic perspectives.
Strengths
- Scientific Relevance: The article tackles a novel and relatively unexplored subject—the role of connexin hemichannels in FSHD—offering valuable insights into complex molecular mechanisms.
- Interdisciplinary Approach: The review integrates knowledge from molecular biology, muscle physiology, and therapeutic approaches, offering a comprehensive view of the pathology.
- Clear Structure: The sections are well delineated and cover introduction, pathogenesis, molecular mechanisms, and therapeutic strategies.
- Use of Figures: The use of illustrations, such as diagrams on molecular mechanisms and disease progression, helps visualize complex concepts.
Criticisms and Suggested Improvements
Molecular Mechanisms
The description of hemichannels is adequate but rather technical. For an interdisciplinary audience, some sections could be simplified, or complex terms could be further explained.
Although the presence of hemichannels in other dystrophies is discussed, the direct connection with FSHD is not always clear, I suggest including (also with an illustration) a conceptual model that links hemichannel activity to the specific muscular degeneration of FSHD.
Therapeutic Perspectives
The section is well-developed, but some strategies (e.g., the use of antioxidants or mimetic peptides) are not deeply contextualized, please discuss the advantage and the limitations of existing strategies.
Literature Review
Many relevant studies are cited, but some key references, such as those on the epigenetic regulation of DUX4, could be explored further. Moreover, some statements (e.g., about the transmission of pathogenic factors through hemichannels) seem speculative and need more robust support from experimental studies.
Specific comments:
It is unclear on what basis the statement *"Are Cx genes direct or indirect targets of DUX4 transcriptional activity? The evidence suggests that they are not, as the data points to a more generalized Cx-related mechanism of skeletal muscle cell death that appears independent of DUX4."* (lines 222–224) relies. Is a reference missing to support this claim?
Could the author explain deeper this statement, the direct or indirect relationship between its transcriptional activity and the influence on Cx genes should be better explained.
Author Response
The description of hemichannels is adequate but rather technical. For an interdisciplinary audience, some sections could be simplified, or complex terms could be further explained.
R: Dear Reviewer, thank you for your valuable suggestion. We think that an image could effectively summarize this section in a clear and concise manner. Instead of adding more text, we have included a simple yet comprehensive image (new Figure 3) that illustrates the role of hemichannels in both physiological and pathological conditions.
Although the presence of hemichannels in other dystrophies is discussed, the direct connection with FSHD is not always clear, I suggest including (also with an illustration) a conceptual model that links hemichannel activity to the specific muscular degeneration of FSHD.
R: Thank you for your valuable suggestion. We added a following text “Since FSHD is driven by the aberrant expression of the DUX4 transcription factor, a critical question is whether Cx genes are direct or indirect targets of DUX4’s transcriptional activity. Current evidence suggests they are not. Skeletal muscle cell death in other pathologies occurs independently of DUX4, and no studies have demonstrated DUX4-mediated regulation of Cx genes in muscle or other tissues. However, an indirect effect remains plausible, as DUX4 induces oxidative stress and inflammation—conditions known to alter connexin expression and function [97–100]. For instance, Cx43 hemichannels are often dysregulated under inflammatory conditions or oxidative stress [74,101,102]. DUX4 also modulates the cellular microenvironment by regulating genes involved in pathways such as inflammation and autophagy, which could indirectly influence connexin expression and function. For example, increased DUX4 expression has been correlated with elevated heme oxygenase-1 (HO-1) levels in hypoxia-resistant cells, and HO-1 activity has been shown to modulate Cx43 and Cx46 hemichannel activity [103]. Regardless of whether DUX4 directly or indirectly regulates Cx expression and function, exaggerated hemichannel activity in dystrophic muscle fibers may increase sarcolemma permeability to critical metabolites like ATP and Ca²⁺. This dysregulation could contribute to cellular dysfunction or death and facilitate the transfer of atrophy-inducing factors—such as calcium ions, free radicals, or extracellular vesicles—between muscle fibers. While sarcolemmal reorganization has been observed in FSHD, its potential association with functional hemichannels remains unexplored, underscoring the need for further investigation (Figure 4)” and a new figure (Figure 4) that illustrate the possible connection between the deregulation of hemichannels and FSHD.
Therapeutic Perspectives
The section is well-developed, but some strategies (e.g., the use of antioxidants or mimetic peptides) are not deeply contextualized, please discuss the advantage and the limitations of existing strategies.
R: We accepted we reviewer suggestion and in the new version we added the following paragraph “In summary, while antioxidants reduce ROS levels, protecting against cellular damage and improving mitochondrial function—which may help preserve muscle fiber integrity—some Cx-based mimetic peptides specifically inhibit hemichannel activity without disrupting gap junction-mediated intercellular communication. By doing so, they have the potential to reduce cell lysis and death while maintaining proper signaling flow between cells. Furthermore, these peptides can significantly decrease extracellular ATP levels and ion dysregulation, leading to reduced muscular inflammation and apoptosis. Their high specificity minimizes off-target effects, enhancing their potential as therapeutic agents. However, these approaches are not without limitations. The effectiveness of antioxidants depends on their ability to penetrate muscle tissues and achieve therapeutic concentrations without triggering pro-oxidant effects, which can arise when ROS levels are excessively suppressed[113,114]. Additionally, long-term antioxidant use may interfere with physiological redox signaling, a process essential for normal cellular function[113,114]. Mimetic peptides, on the other hand, face challenges related to stability, delivery, and bioavailability[115]. These peptides often require advanced delivery systems to reach their targets effectively, and their susceptibility to enzymatic degradation in the bloodstream can further reduce therapeutic efficacy. Moreover, the heterogeneity of FSHD pathophysiology means that these treatments may address only specific aspects of the disease, such as oxidative stress or hemichannel dysfunction, without addressing the underlying causes of muscle degeneration. Recognizing these limitations underscores the need for further research aimed at optimizing the delivery, efficacy, and safety of antioxidants and mimetic peptides. Combining these approaches with other therapeutic strategies could provide a more comprehensive and effective solution for managing the multifaceted nature of FSHD.”
Literature Review
Many relevant studies are cited, but some key references, such as those on the epigenetic regulation of DUX4, could be explored further. Moreover, some statements (e.g., about the transmission of pathogenic factors through hemichannels) seem speculative and need more robust support from experimental studies.
R: In this revised version, we have incorporated many new references to address the issue raised by the reviewer.
Specific comments:
It is unclear on what basis the statement *"Are Cx genes direct or indirect targets of DUX4 transcriptional activity? The evidence suggests that they are not, as the data points to a more generalized Cx-related mechanism of skeletal muscle cell death that appears independent of DUX4."* (lines 222–224) relies. Is a reference missing to support this claim? Could the author explain deeper this statement, the direct or indirect relationship between its transcriptional activity and the influence on Cx genes should be better explained.
R: We hope this new paragraph fulfil the doubts of the revisor; “Since FSHD is driven by the aberrant expression of the DUX4 transcription factor, a key question is whether Cx genes are direct or indirect targets of DUX4's transcriptional activity. The available evidence suggests they are not, as other skeletal muscle cell death in other skeletal muscle pathologies is independent of DUX4, additionally, in the literature there is no evidence of a DUX4-mediated Cx-gene regulation in muscles or any other tissue. However, an indirect effect remains plausible, as DUX4 induces oxidative stress and inflammation—conditions known to alter connexin expression and function[91–94]. For example, Cx43 hemichannels are often dysregulated in inflammatory states or under oxidative stress[68,95,96]. Additionally, DUX4 influences the cellular microenvironment by activating or repressing genes associated with key signaling pathways, such as those involved in inflammation or autophagy, which could indirectly affect Cx expression and function. Thus for example, an increase in DUX4 has been correlated with elevated heme oxygenase-1 (HO-1) expression in hypoxia-resistant cells, and HO-1 activity has been shown to modulate Cx43 and Cx46 hemichannel activity[97]. Whether or not Cx expression and function are regulated by DUX4, it can be suggested that in dystrophic muscle fibers, the presence of hemichannels with exaggerated activity may increase sarcolemma permeability to critical metabolites such as ATP and Ca2+. This dysregulation could contribute to cell dysfunction or even cell death, and also facilitate the transfer of atrophy-inducing factors between muscle fibers. Interestingly, while sarcolemmal reorganization has been observed in FSHD, it has not yet been linked to the presence of functional hemichannels, highlighting the need for further investigation.”